# Chemical inhibition of the integrated stress response impairs the ubiquitin-proteasome system
Shanshan Xu[1], Maria E. Gierisch[1], Enrica Barchi[1], Ina Poser[2,4], Simon Alberti[2,3], Florian A. Salomons[1] & Nico P. Dantuma [1] ✉

Inhibitors of the integrated stress response (ISR) have been used to explore the potential beneficial effects of reducing the activation of this pathway in diseases. As the ISR is in essence a protective response, there is, however, a risk that inhibition may compromise the cell's ability to restore protein homeostasis. Here, we show that the experimental compound ISRIB impairs degradation of proteins by the ubiquitin-proteasome system (UPS) during proteotoxic stress in the cytosolic, but not nuclear, compartment. Accumulation of a UPS reporter substrate that is intercepted by ribosome quality control was comparable to the level observed after blocking the UPS with a proteasome inhibitor. Consistent with impairment of the cytosolic UPS, ISRIB treatment caused an accumulation of polyubiquitylated and detergent insoluble defective ribosome products (DRiPs) in the presence of puromycin. Our data suggest that the persistent protein translation during proteotoxic stress in the absence of a functional ISR increases the pool of DRiPs, thereby hindering the efficient clearance of cytosolic substrates by the UPS.

The integrated stress response (ISR) is a conserved signaling network that couples cellular stress detection to protein synthesis with the final goal of maintaining protein homeostasis[1]. Extracellular and intracellular stress conditions can activate the ISR through stress-sensing kinases that phosphorylate serine residue 51 of the eukaryotic initiation factor 2α (eIF2α) subunit[2,3]. Eukaryotic initiation factor 2 (eIF2) is a heterotrimeric GTPase composed of α, β, and γ subunits, which form a ternary complex with GTP and the methionine initiator tRNA to initiate mRNA translation[4]. Once the start codon is decoded, GTP is hydrolyzed to GDP by the guanine nucleotide exchanging factor eIF2B, and eIF2 is released from the ribosome, forming a new ternary complex to engage in another round of translation initiation[5]. Phosphorylated eIF2α inhibits the nucleotide exchanging activity of eIF2B, thereby limiting the recycling of eIF2, which results in reduced translation initiation and protein synthesis[6]. The untranslated mRNA in complex with disassembled 40S ribosome subunits interacts with RNA binding proteins and undergoes phase separation giving rise to cytosolic stress granules, which are membrane-less structures that function as temporary storage deposits for untranslated mRNAs[7].

While the ISR is essentially cytoprotective and increases cell viability and survival during challenging conditions, its activation has also been linked to various pathological conditions, including cancer, diabetes, and neurodegenerative diseases[1]. The prevalence of ISR activation in these diseases has motivated the search for drugs that can inhibit or block this signaling cascade and prevent possible damage caused by the aberrant induction of this response. The Integrated Stress Response Inhibitor (ISRIB) is an experimental compound originally identified in a high-throughput screen for inhibitors that selectively mitigate PERK signaling[6]. Further studies revealed that ISRIB inhibits the ISR by allosterically destabilizing the complex between eIF2B and phosphorylated eIF2α[8]. By preventing the inhibitory effect of phosphorylated eIF2α on eIF2B, ISRIB blunts the ISR's ability to block protein synthesis and induce the formation of stress granules[9]. Thus, in the presence of ISRIB, protein synthesis persists, and consequently, stress granules are not formed in response to proteotoxic stress.

It has been proposed that pharmacological inhibition of the ISR may be beneficial in the context of neurodegenerative diseases that are linked to the activation of this pathway[10]. Indeed, ISRIB has been shown to enhance cognitive memory in prion-diseased mice[11]. Considering the protective nature of the ISR, adverse effects may occur, but there is currently little insight into the negative consequences of curtailing this response.

[1]Department of Cell and Molecular Biology (CMB), Karolinska Institutet, Solnavägen 9, S-17165 Stockholm, Sweden. [2]Max Planck Institute of Molecular Cell Biology and Genetics, Dresden, Germany. [3]Biotechnology Center (BIOTEC), Center for Molecular and Cellular Bioengineering (CMCB), Technische Universität Dresden, Dresden, Germany. [4]Present address: Open Sesame Therapeutics GmbH, Pfotenhauerstr. 108, 01307 Dresden, Germany. ✉e-mail: nico.dantuma@ki.se

Induction of ER stress sensitizes cancer cells to ISRIB toxicity, consistent with a compromised ability of cells to deal with stress conditions[6,12]. This synergistic effect may allow selective targeting of cancer cells[13], but it may also undermine its applicability in neurodegenerative diseases as it could exacerbate cellular pathologies that involve ER stress[14]. Moreover, other studies have reported that prolonging the ISR has a therapeutic effect in mouse models for protein-misfolding diseases[15–17]. While these findings are not mutually exclusive, they raise questions on how inhibition of the ISR affects other protein quality control mechanisms in the cell.

The ubiquitin-proteasome system (UPS) is the primary pathway for the elimination of misfolded proteins during proteotoxic stress[18,19]. In essence it consists of a two-step process in which misfolded proteins are first modified by the small protein modifier ubiquitin and next degraded by the proteasome complex, which selectively hydrolyzes ubiquitylated proteins[20]. While the ISR aims at restoring proteostasis by preventing the synthesis of misfolded proteins, the UPS contributes to this effort through the elimination of already existing misfolded proteins through proteolytic destruction. Thus, these two processes join efforts in restoring and maintaining protein homeostasis by reducing protein synthesis and stimulating degradation[21]. This calls into question whether the persistent protein synthesis in ISRIB-treated cells challenges the UPS during proteotoxic stress by increasing the load of misfolded substrates.

Here, we report that ISRIB treatment negatively affects the UPS and compromises its ability to clear ubiquitylated substrates in cells after a proteotoxic stress insult. Consistent with a causal link with protein synthesis, ISRIB affects primarily the UPS in the cytosolic compartment and causes an increase in polyubiquitylated and insoluble defective newly synthesized proteins (DRiPs). Our data suggest that the persistent synthesis of proteins under conditions that compromise protein folding impairs the UPS and may thereby generate an intracellular environment that favors protein aggregation, a condition intrinsically connected to age-related neurodegenerative diseases.

## Results

### ISRIB aggravates UPS impairment during proteotoxic stress

We have previously shown that the induction of proteotoxic stress by thermal stress transiently impairs the functionality of the UPS, resulting in the accumulation of polyubiquitylated substrates[22]. In this study, we used a mild heat shock as an experimental paradigm for an instant, reversible proteotoxic stress that induces the ISR[23–25], including the formation of cytosolic stress granules[26]. To simultaneously access the activation of the ISR and the functional status of the UPS, we generated a human melanoma MelJuSo cell line that stably expresses mCherry-tagged G3BP1, a marker for stress granules[27], and the ubiquitin fusion degradation substrate ubiquitin$^{G76V}$-yellow fluorescent protein (Ub-YFP), a sensor for ubiquitin-dependent proteasomal degradation[28]. Exposure of the reporter cell line to a mild heat shock induced the formation of cytosolic stress granules, as evidenced by the punctate cytosolic localization of mCherry-G3BP1 (Fig. 1a, upper panels). Heat shock-induced expression of the ISR transcription factor ATF4 (Suppl. Fig. 1a) and induced phosphorylation of eIF2α (Suppl. Fig. 1b), as reported previously[25]. Phosphorylation of eIF2α was observed upon heat shock in the absence or presence of ISRIB (Suppl. Fig. 1b), consistent with ISRIB inhibiting the ISR downstream of phosphorylated eIF2α[8]. Importantly, the induction of stress granules, monitored by the formation of mCherry-G3BP1 puncta, was blocked by ISRIB treatment (Fig. 1a, upper panels), confirming efficient inhibition of the ISR. The inclusion of ISRIB during the mild heat shock was well tolerated and did not have a detectable effect on cell viability during the recovery phase (Suppl. Fig. 1c).

The functionality of the UPS was assessed by analyzing the levels of the Ub-YFP reporter substrate in cells that had been thermally stressed in the absence or presence of ISRIB followed by a recovery phase without ISRIB. Ub-YFP accumulated in the aftermath of the proteotoxic insult, indicating inefficient clearance of ubiquitylated proteins by proteasomal degradation in untreated and ISRIB-treated, thermally stressed cells

(Fig. 1a, lower panels). Accumulation of Ub-YFP in response to heat shock is consistent with our earlier finding that mild thermal stress has a negative impact on ubiquitin-dependent proteasomal degradation[22]. Notably, quantitative analysis of the Ub-YFP levels 4 h after heat shock showed that ISRIB treatment caused a significant increase in the accumulation of the Ub-YFP reporter as compared to untreated, thermally stressed cells (Fig. 1b). The increase in ISRIB-treated cells corresponded to approximately 10% of the increase in the Ub-YFP levels that was obtained when the UPS was fully blocked with 200 nM of the proteasome inhibitor epoxomicin (Suppl. Fig. 2). A time-course analysis confirmed that the Ub-YFP accumulated to higher levels in ISRIB-treated as compared to untreated cells with a maximal difference occurring 4–5 h into the recovery phase after which the Ub-YFP levels started to gradually decline reaching basal levels after 14 h (Fig. 1c, d).

Pulse-chase experiments confirmed a delayed clearance of the Ub-YFP reporter in ISRIB-treated cells that recovered from thermal stress, supporting the model that the elevated steady-state levels are a consequence of slower degradation of Ub-YFP during the recovery phase (Fig. 1e). Although we cannot exclude a contribution of persistent protein synthesis to the elevated Ub-YFP levels in ISRIB-treated cells, the labeled Ub-YFP levels at the start of the recovery phase were comparable, suggesting that ISRIB did not substantially affect the synthesis of the reporter during the heat shock. The strongest effect on the turnover of Ub-YFP was observed during the initial stage after the heat shock and preceded the time point at which maximal accumulation of the reporter occurred. This suggests that degradation is mostly affected during or shortly after the stress response, followed by a short delay, after which the reporter levels gradually start to build up.

Impairment of the UPS in cells recovering from thermal stress is confined to ubiquitin-dependent substrates[22]. To test if the same applies to cells exposed to ISRIB, we analyzed the steady-state levels of a green fluorescent protein (GFP) reporter containing the degradation signal of ornithine decarboxylase (ODC), a ubiquitin-independent substrate of the UPS[29]. The GFP-ODC levels were not increased in thermally stressed cells regardless of the absence or presence of ISRIB, whereas inhibition of the proteasome by epoxomicin caused accumulation of this substrate (Fig. 1f). This shows that the effect of ISRIB is confined to the degradation of ubiquitylated substrates and suggests that there is sufficient proteasome activity in ISRIB-treated cells to execute degradation. Consistent with the preserved proteasome function, ISRIB did not have a significant effect on the enzymatic activity of proteasomes in thermally stressed cells (Suppl. Fig. 3a). Moreover, ISRIB treatment did not result in a detectable increase in their sensitivity to inhibition of the proteasome by epoxomicin (Suppl. Fig. 3b).

### Inhibition of ISR is responsible for UPS dysfunction in stressed cells

In order to determine if the aggravation of the UPS impairment in ISRIB-treated cells is indeed caused by its ability to inhibit the ISR, we compared the effect of ISRIB on the UPS with another compound that inhibits the ISR by an unrelated mechanism. GSK2606414 is an inhibitor of PERK, which has been shown to be responsible for heat shock-induced activation of the ISR by phosphorylating eIF2α[23–25]. Side-by-side comparison of the effect of administration of ISRIB and GSK2606414 on the Ub-YFP reporter levels after heat shock showed that both compounds caused a very similar accumulation of the substrate, consistent with inhibition of the ISR being responsible for the impaired UPS (Fig. 2a).

To assess whether the same holds true for other proteotoxic insults that induce the ISR, we analyzed the effect of ISRIB on the functionality of the UPS upon induction of oxidative stress by exposing cells to sodium arsenate. Treatment for 4 h to 50 μM sodium arsenate induced the formation of stress granules in the Ub-YFP reporter cells, which was efficiently blocked by co-administration of ISRIB (Fig. 2b). As observed following thermal stress, the administration of ISRIB caused a larger increase in the steady-state levels of the Ub-YFP in response to sodium arsenate (Fig. 2c, d), suggesting that the accumulation of UPS substrates is a general consequence of ISR inhibition during proteotoxic stress.

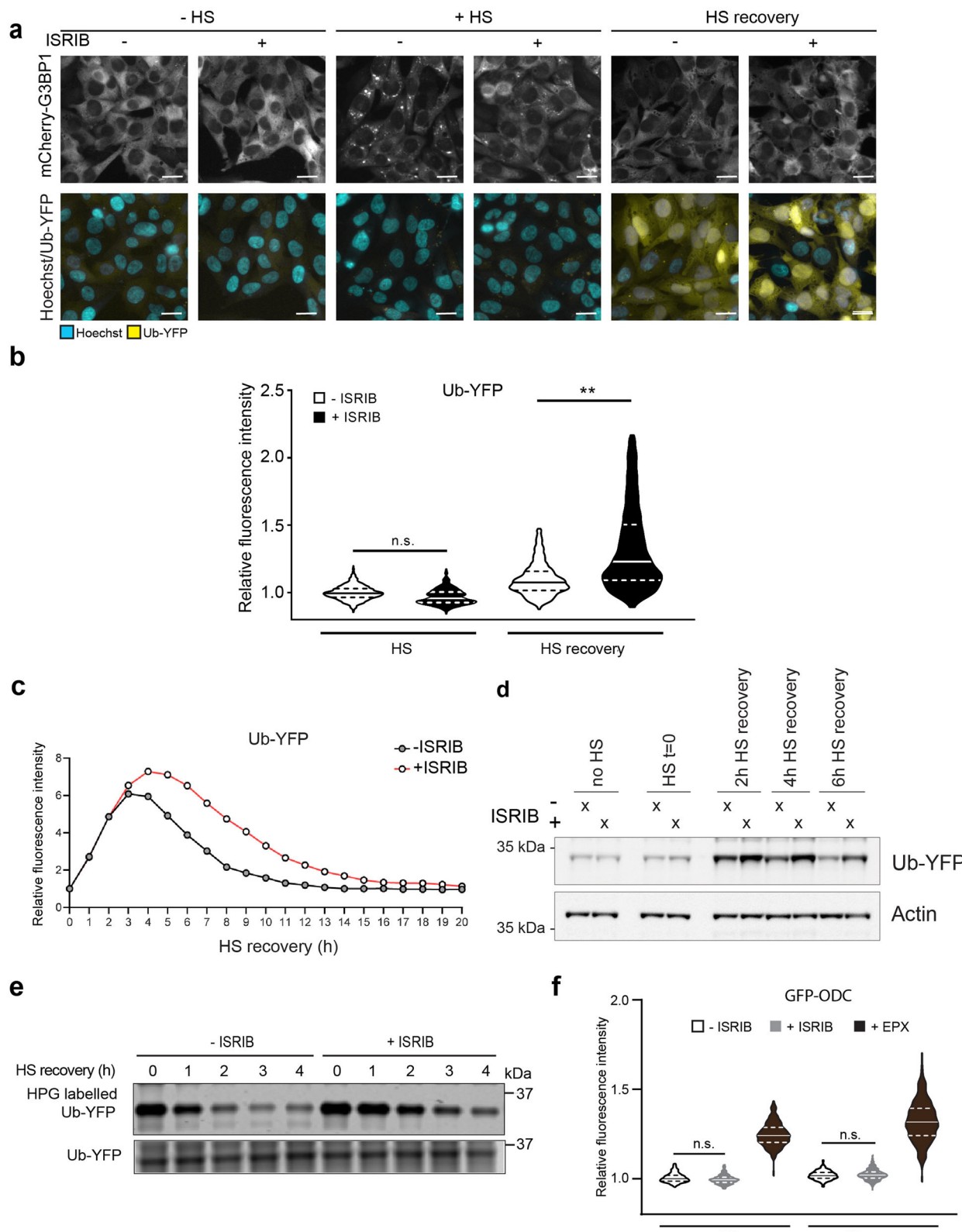

## ISRIB causes accumulation and sequestration of proteasome substrates in the cytosolic, but not nuclear, compartment

We have shown previously that genetic inhibition of ISR-induced formation of stress granules primarily affects the functionality of the UPS in the nuclear compartment with little effect on the UPS in the cytoplasm[30]. As ISRIB also prevents the formation of stress granules, we wondered if chemical interference with the formation of stress granules had a similar effect on the nuclear UPS. To address this question, we analyzed the UPS in the nuclear and cytosolic compartments using two compartment-specific proteasome substrates: NLS-GFP-NLS-GFP-CL1 (referred to as NLS-GFP-CL1) and NES-GFP-NES-GFP-CL1 (referred to as NES-GFP-CL1)[31]. The 16 amino acid-long CL1 extension behaves as an aggregation-prone degradation

**Fig. 1 | ISRIB aggravates UPS impairment during proteotoxic stress.**
**a** Representative micrographs of the effect of ISRIB on Ub-YFP in thermally stressed cells. Cells were left untreated (-ISRIB) or pretreated with ISRIB (+ISRIB) for 30 min. After this initial incubation, they were left untreated (−HS) or exposed to 43 °C for 30 min (+HS) and followed for 4 h after the heat shock in the absence of ISRIB (HS recovery). Images were captured with a wide-field, high-content microscope. Scale bar, 20 μm. **b** Quantification of mean cellular YFP fluorescence intensities per cell. Experimental setup as described in (**a**). Values were normalized to untreated control cells (-ISRIB, −HS). The frequency and distribution of the relative fluorescence intensities per cell are shown as violin plots. The solid lines in each distribution represent the medians, and the dash lines represent the upper and lower interquartile range limits (three independent experiments, >1000 cells analyzed per condition, Kruskal–Wallis test, **$P < 0.01$, n.s.: not significant). **c** Effect of ISRIB on accumulation kinetics of Ub-YFP. Kinetics of Ub-YFP accumulation in control (-ISRIB) and ISRIB-treated (+ISRIB) MelJuSo cells, in response to a 30-min 43 °C heat shock, followed by 14 h at 37 °C (HS recovery). Data points show the mean fluorescence intensity per image. Four sites were imaged, per treatment, per time point. Error bars indicate the standard deviation

between the four images taken for the same condition (time and treatment). Here we show one representative experiment out of two independent replicates. **d** Effect of ISRIB on Ub-YFP accumulation in stressed cells. MelJuSo cells expressing Ub-YFP were incubated in the absence (-ISRIB) or presence of ISRIB (+ISRIB) for 30 min, before being subjected to 43 °C heat shock for 30 min. The Ub-YFP levels were followed throughout the recovery phase (HS recovery). **e** Pulse-chase experiment with HPG-labeled Ub-YFP. The upper panel is an in-gel detection of HPG labeled Ub-YFP in a pulse-chase experiment during the recovery phase, in the absence (-ISRIB) or presence of ISRIB (+ISRIB). The total Ub-YFP reporter levels were detected in the lysate by immunoblotting (lower panel). A representative result of three independent experiments is shown. **f** Effect of ISRIB on ubiquitin-independent proteasomal degradation. Quantification of GFP intensities of MelJuSo cells expressing GFP-ODC. Cells were left untreated (-ISRIB) or incubated with ISRIB (+ISRIB) for 30 min before being left untreated (37 °C) or exposed to a 43 °C heat shock for 30 min, followed by recovery (HS recovery) for 4 h. Values were normalized to untreated control cells (-ISRIB, −HS). The violin plots represent the frequency and distribution of the relative fluorescence intensities per cell, quantified as nuclear YFP intensity.

signal[32], resulting in accumulation of the reporter in intracellular aggregates if not efficiently cleared[28,33]. Surprisingly, ISRIB pre-treatment did not affect the degradation of the nuclear reporter in thermally stressed cells (Fig. 3a, b), but it inhibited the degradation of the cytosolic reporter (Fig. 3c, d). The stabilization of the cytosolic aggregation-prone reporter was accompanied by an increase in the number of cells displaying accumulation of the reporter in the perinuclear region (Fig. 3c, e), a typical localization for sequestration of cytosolic protein aggregates[34,35]. Thus, in contrast to the nuclear UPS dysfunction in stress granule-deficient cells, the combined action of ISRIB on stress granule formation and protein synthesis results in the impairment of ubiquitin-dependent proteasomal degradation in the cytosolic compartment.

## ISRIB impairs the degradation of DRiPs

As ISRIB prevents the stress-induced downregulation of protein synthesis[36], we reasoned that treatment with this compound under proteotoxic stress conditions may increase the levels of DRiPs as proper co-translational folding of nascent polypeptides may become problematic. To address this question, we labeled the pool of newly synthesized peptides produced during thermal stress with puromycin. As incorporation of puromycin will terminate protein translation, this will give rise to truncated polypeptides that are commonly referred to as defective ribosome products (DRiPs), which are targeted for proteasomal degradation[37]. As anticipated, based on the biological activity of ISRIB, the decrease in protein synthesis induced by thermal stress was partially reversed by ISRIB treatment (Fig. 4a). In untreated thermally stressed cells, DRiPs were enriched at TIA1-positive stress granules in the cytosol and displayed, in addition, a nuclear localization (Fig. 4b, upper panels). The nuclear localization is consistent with two earlier studies that show that DRiPs passively diffuse into the nucleus and suggest an important role of nuclear protein quality control in handling these products[30,38]. In ISRIB-treated cells, nuclei were, however, largely devoid of DRiPs, and instead, a prominent accumulation of DRiPs was observed throughout the cytosol (Fig. 4b, lower panels).

To investigate the levels of ubiquitylated DRiPs, Tandem Ubiquitin Binding Entities (TUBEs) were used to pulldown polyubiquitylated, puromycin-labeled proteins from the lysate of cells exposed to heat shock in the absence or presence of ISRIB. Indeed, ISRIB treatment promoted the accumulation of ubiquitylated DRiPs in stressed cells (Fig. 4c). In line with the notion that the persistent translation under stress conditions increases the load of misfolded or unfolded proteasome substrates, the administration of ISRIB to cells exposed to heat shock increased the pool of Triton X-100 insoluble ubiquitylated proteins (Fig. 4d). These results indicate that persistent protein synthesis during proteotoxic stress causes a cytosolic accumulation of ubiquitylated proteins that are prone to aggregation.

## ISRIB impairs degradation of an RQC reporter substrate

As DRiPs are typically targeted for proteasomal degradation in the cytosol by the ribosome quality control (RQC) machinery[39], we wondered if the

increase in DRiPs in ISRIB-treated cells had an impact on the ability of cells to efficiently clear RQC substrates. We first checked if RQC is affected by ISRIB by analyzing the effect of ISRIB treatment on ribosome stalling at poly(A) sequences using a poly(A) readthrough reporter[40]. This showed that ISRIB treatment did not interfere with the initial steps of the RQC, leading to persistent ribosome stalling and disassembly since the ribosomes did not readthrough the poly(A) sequence (Fig. 5a, b). To analyze the fate of polypeptides intercepted by RQC, we used a GFP^nonstop reporter, that due to a mutation of the GFP stop codon, produces a GFP-tagged readthrough RQC substrate that can be readily quantified[41]. We found that ISRIB caused an increase in GFP^nonstop levels that was comparable with the levels reached upon treatment with the proteasome inhibitor epoxomicin (Fig. 5c, d). Co-treatment of ISRIB and epoxomicin did not cause a further accumulation of GFP^nonstop, indicating that the accumulation of GFP^nonstop in stressed, ISRIB-treated cells was not due to an increase in synthesis of the reporter but caused by a delayed clearance of GFP^nonstop by the UPS. Thus, the increase in DRiPs is accompanied by inefficient clearance of RQC substrates by the UPS.

## Discussion

In summary, we show that inhibition of the ISR compromises ubiquitin-dependent proteasomal degradation in cells that have been exposed to a proteotoxic insult. Our data point to the persistent protein synthesis during proteotoxic stress as the primary cause of this effect, as ISRIB blunts the protective inhibition of protein synthesis, which is normally triggered by the ISR[42]. The reduction of protein synthesis during proteotoxic stress is best understood as an attempt to minimize the accumulation of aggregation-prone proteins, as nascent chains will be hindered from adopting their native conformation under challenging conditions[42]. Chemical inhibition of the ISR is likely to increase the load of newly synthesized proteins that need to be targeted for proteasomal degradation. The partial increase in the levels of newly synthesized proteins that we observed in ISRIB-treated stressed cells is likely to be an underestimation as a large pool of these proteins may be co-translationally degraded by the RQC pathway.

Our data indicate that in ISRIB-treated cells, degradation of ubiquitin-dependent, but not ubiquitin-independent, substrates is compromised. We have previously shown that impairment of the UPS during proteotoxic stress is not caused by saturation of the capacity of proteasomal activity in cells[22]. Instead, the available pool of free ubiquitin appears to be a limiting factor, which gives rise to competition between substrates for ubiquitin tagging, a demand that will dramatically increase during proteotoxic stress due to the accumulation of misfolded proteins[22]. Our finding that the activity of the proteasome is not reduced in ISRIB-treated cells, which is also supported by the unabated degradation of a ubiquitin-independent substrate, is consistent with an increased demand on ubiquitylation as the primary cause for the accumulation of UPS substrates. We have speculated that the competition for ubiquitin may be important to provide crosstalk

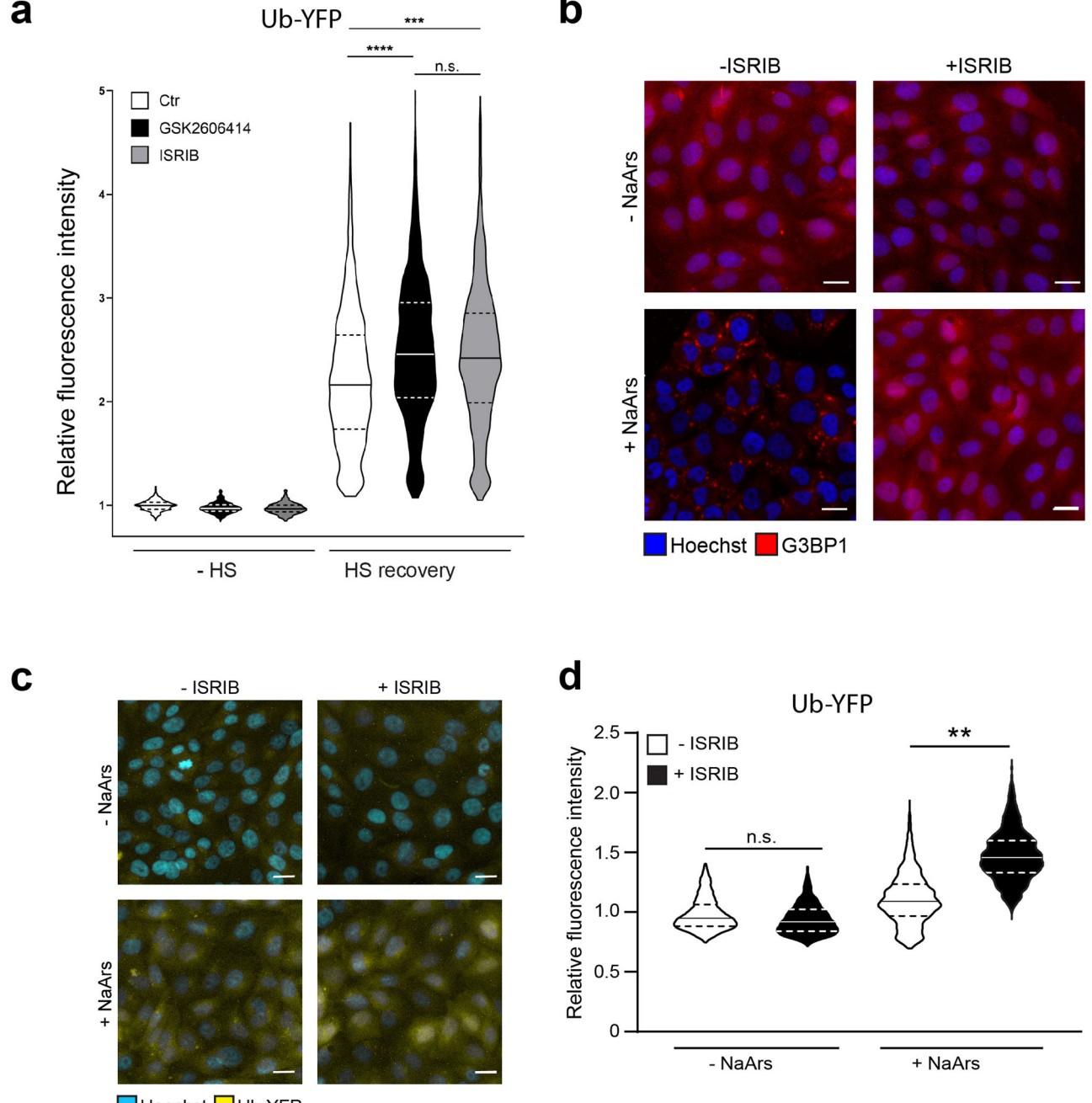

**Fig. 2 | Inhibition of the ISR is responsible for UPS dysfunction in stressed cells.**
**a** Comparison of the effects of ISRIB and GSK2606414 on Ub-YFP reporter's accumulation, either without heat shock (−HS) or with 30 min at 43 °C, followed by 2 h of recovery (HS recovery). Control cells were left untreated (Ctr), while ISRIB and GSK2606414 were administered to the cells 30 min ahead of heat shock (or regular 37 °C incubation for the −HS samples). Values were normalized to untreated control cells (Ctr, −HS). The violin plots represent the frequency and distribution of the relative fluorescence intensities per cell, quantified as nuclear YFP intensity. The solid lines in each distribution represent the medians, and the dash lines represent the upper and lower interquartile range limits (one out of two representative experiments, >300 cells analyzed per condition, Kruskal–Wallis test, ****$P < 0.0001$, *** $P < 0.001$, n.s.: not significant). **b** Representative micrographs of MelJuSo cells expressing G3BP1-mCherry that were either pre-treated in the absence (-ISRIB) or presence (+ISRIB) of

ISRIB for 30 min. After pre-treatment, cells were treated with or without sodium arsenate (±NaArs) for 4 h. Scare bar, 20 μm. **c** Representative micrographs of the effect of ISRIB on Ub-YFP in NaArs-stressed cells. Cells were left untreated (-ISRIB) or pretreated with ISRIB (+ISRIB) for 30 min. After this initial incubation they were left untreated (-NaArs) or exposed to sodium arsenate for 4 h (+NaArs). Images were captured with a wide-field, high-content microscope. Scale bar, 20 μm. **d** Effect of ISRIB on Ub-YFP reporter levels in NaArs-stressed cells. Quantification of mean cellular YFP fluorescence intensities. Experimental setup as described in (**d**). Values were normalized to untreated control cells (-ISRIB, −HS). The frequency and distribution of the relative fluorescence intensities per cell are shown as violin plots. The solid lines in each distribution represent the medians, and the dash lines represent the upper and lower interquartile range limits (three independent experiments, >1000 cells analyzed per condition, Kruskal–Wallis test, ** $P < 0.01$).

between the various proteolytic and non-proteolytic processes that critically depend on ubiquitin conjugation[43,44]. It is, therefore, possible that ISRIB will also indirectly compromise other ubiquitin-dependent processes, which may have a further negative impact on the ability of cells to cope with stress

conditions. As the potential negative consequences on these cellular processes may become a source for adverse effects, it will be important to further explore the effects of ISRIB and other ISR inhibitors on other ubiquitin-dependent processes.

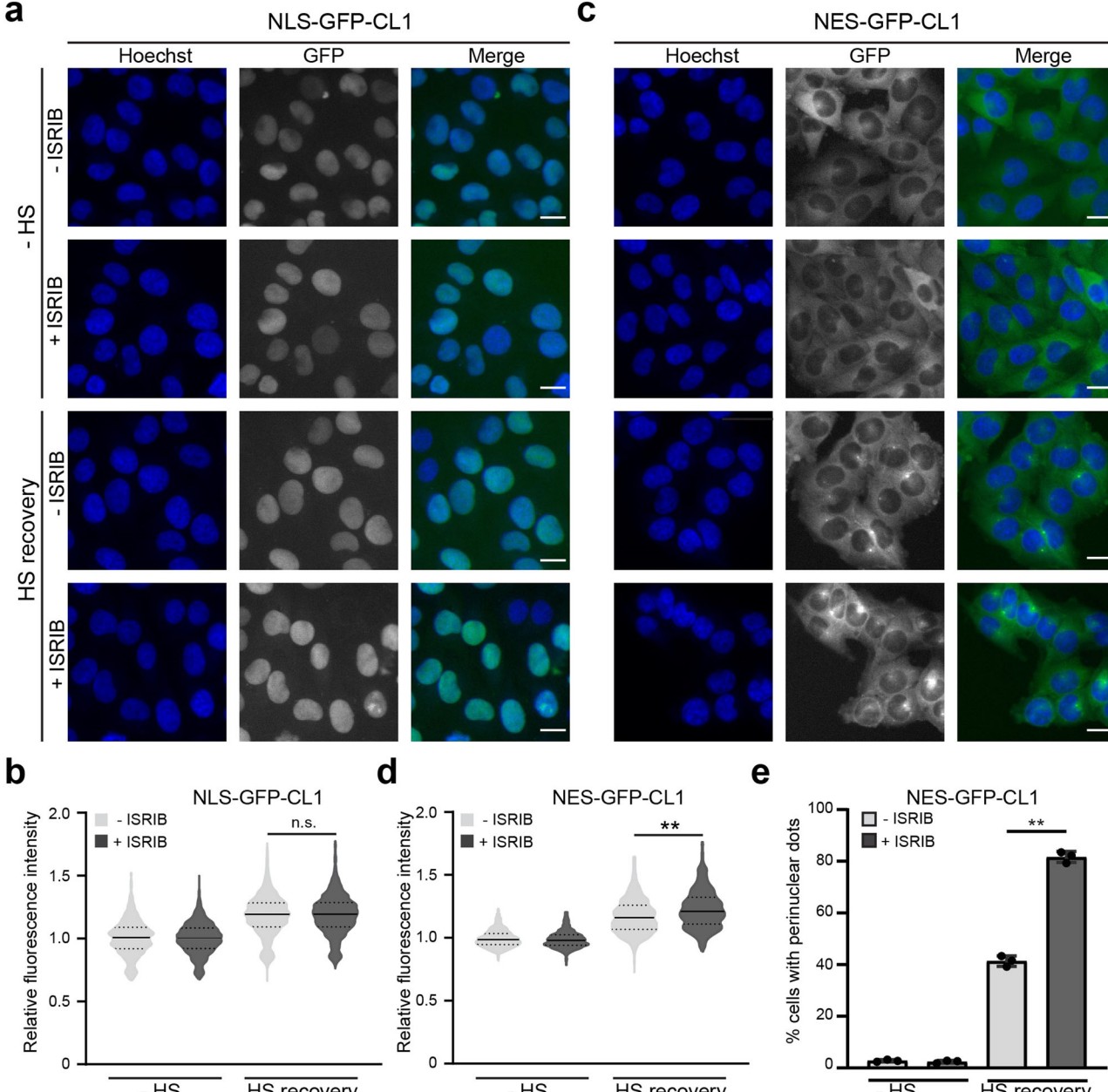

**Fig. 3 | ISRIB causes accumulation and sequestration of proteasome substrates in the cytosolic, but not nuclear, compartment. a** Effect of ISRIB on nuclear UPS. Representative micrographs of MelJuSo cells expressing NLS-GFP-CL1. Cells were left untreated (-ISRIB) or incubated with ISRIB (+ISRIB) for 30 min before being left untreated (−HS) or exposed to a 43 °C heat shock for 30 min and followed by recovery (HS recovery) for 4 h. Images were captured with a wide-field, high-content microscope. Scale bar, 20 μm. **b** Quantification of mean cellular NLS-GFP-CL1 fluorescence intensities from experiment shown in (**a**). Values were normalized to untreated control cells (-ISRIB, −HS). The frequency and distribution of the relative fluorescence intensities per cell are shown as violin plots (three independent experiments, >1000 cells analyzed per condition, Kruskal–Wallis test, n.s.: not significant). **c** Effect of ISRIB on cytosolic UPS. Representative micrographs of MelJuSo cells expressing NES-GFP-CL1. Cells were left untreated (-ISRIB) or incubated with ISRIB (+ISRIB) for 30 min before being left untreated (−HS) or exposed to a 43 °C heat shock for 30 min and followed by recovery (HS recovery) for 4 h. Images were captured with a wide-field, high-content microscope. Scale bar, 20 μm. **d** Quantification of mean cellular NES-GFP-CL1 fluorescence intensities from experiment shown in (**c**). Values were normalized to untreated control cells (-ISRIB, −HS). The frequency and distribution of the relative fluorescence intensities per cell are shown as violin plots. (three independent experiments, >1000 cells analyzed per condition, Kruskal–Wallis test, **$P < 0.01$). **e** Percentage of cells that are positive for cytosolic NES-GFP-CL1 foci from experiment shown in (**c**). Data represent the mean ± SD. (three independent experiments, Kruskal–Wallis test, **$P < 0.01$).

Interestingly, ISRIB compromised the degradation of cytosolic substrates with no detectable effects on ubiquitin-dependent proteasomal degradation in the nuclear compartment. As ubiquitin molecules freely diffuse between the cytosolic and nuclear compartments[43], this suggests that depletion of the pool of free ubiquitin in thermally stressed cells[22], is unlikely to be the sole explanation for the aggravated UPS impairment by ISRIB. The general effect on ubiquitin-dependent proteasomal degradation in the cytosolic compartment may also explain the original observation that ISRIB treatment compromises the ability of cells to cope with endoplasmic reticulum (ER) stress[6], as dealing with this condition requires efficient degradation of misfolded ER proteins through ubiquitin-dependent ER-associated degradation (ERAD) by the UPS[45]. Notably, ER stress itself has already been found to push the UPS to its limits[28], and further havoc of the UPS may be the reason for the impaired ability of ISRIB-treated cells to adapt to ER stress.

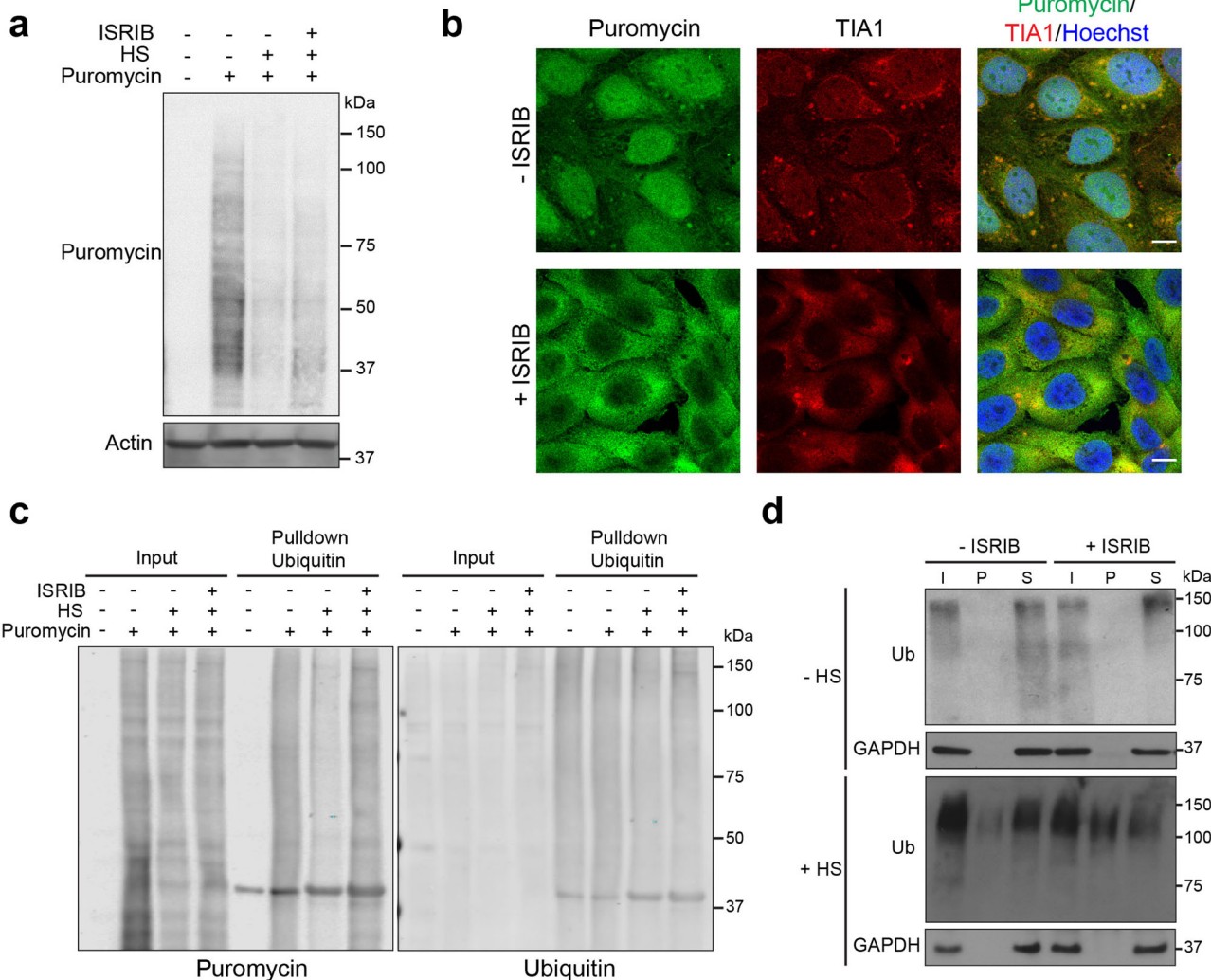

**Fig. 4 | ISRIB impairs the degradation of DRiPs. a** Effect of ISRIB on DRiP synthesis. Analysis of puromycin incorporation in MelJuSo cells pre-treated without (-ISRIB) or with ISRIB (+ISRIB). Cells were left untreated (- HS) or exposed to 43 °C for 30 min (+HS). **b** Effect of ISRIB on localization of DRiPs. Representative micrographs of MelJuSo cells that were pre-treated with (+ISRIB) or without ISRIB (-ISRIB). Cells were exposed to 43 °C for 30 min (+HS) in the presence of puromycin. Puromycin and the stress granule marker TIA1 were visualized by immunostaining. Images were captured with a laser scanning confocal microscope. Scare bar, 10 μm. **c** Western blot analysis of ubiquitylated DRiPs. MelJuSo cells expressing Ub-YFP that were incubated in the absence of presence of ISRIB for

30 min before left untreated or subjected to 43 °C heat shock for 30 min with or without 5 ug/ml puromycin. The cells were lysed as indicated and ubiquitylated proteins were pulled down with tandem ubiquitin binding entities (TUBEs). The input and pulldown samples were analyzed by immunoblotting with antibodies directed against ubiquitin and puromycin. **d** Western blot analysis of Triton X-100 solubility assay of MelJuSo cells incubated with or without ISRIB for 30 min before left untreated (−HS) or subjected to 43 °C heat shock for 30 min (+HS). Ubiquitylated proteins were detected in total lysates (I), insoluble pellet fraction (P), and soluble fraction (S). GAPDH is used as a control for soluble proteins.

The ISR appears to be a double-edged sword as its intrinsic protective effect on the detrimental consequence of proteotoxic stress goes hand in hand with its contribution to the etiology of a number of diseases that are associated with aberrant activation of this stress reponse[1]. In support of the overall beneficial effect of ISRIB, it has been shown that ISRIB can readily cross the blood-brain barrier with good pharmacokinetic properties and enhance long-term memory in mice[6]. Moreover, the capacity of ISRIB to prevent the formation of stress granules may have additional therapeutic potential since these structures have been linked to several neurodegenerative diseases, most notably ALS[9]. Yet, the effect of ISRIB administration on ALS pathology in mouse models has been puzzling as both improvement and worsening of the symptoms have been reported[46,47]. On the other hand, prolonging the ISR by pharmacological inhibition of the dephosphorylation of eIF2α, which results in a delay in translational recovery, prevents physiological and molecular defects in mouse models for Charcot-Marie-Tooth 1B, ALS, and Huntington's disease, suggesting general beneficial effects in

protein-misfolding diseases[15–17]. Maintaining efficient protein quality control by preserving UPS activity through extending the time window for ISR-induced translational inhibition may contribute to the protective effect of these compounds.

Whether the effect of ISRIB and other ISR inhibitors will be beneficial or detrimental may depend on the ability of such compounds to prevent the aberrant activation of the ISR without hindering its protective function. Earlier studies suggest that ISRIB may meet this requirement as the robust activation of the ISR in response to severe stress, is not compromised by ISRIB, whereas the lingering aberrant response associated with pathologies is efficiently blunted[36,48]. Our data suggest that, even though the use of ISRIB may be promising in therapeutic settings, caution should be practiced as its negative impact on the clearance of misfolded proteins by the UPS may exacerbate conditions that favor protein aggregation. In particular, chronic curtailing of the ISR in neurons can turn out to be problematic since the accompanying UPS impairment may accelerate the

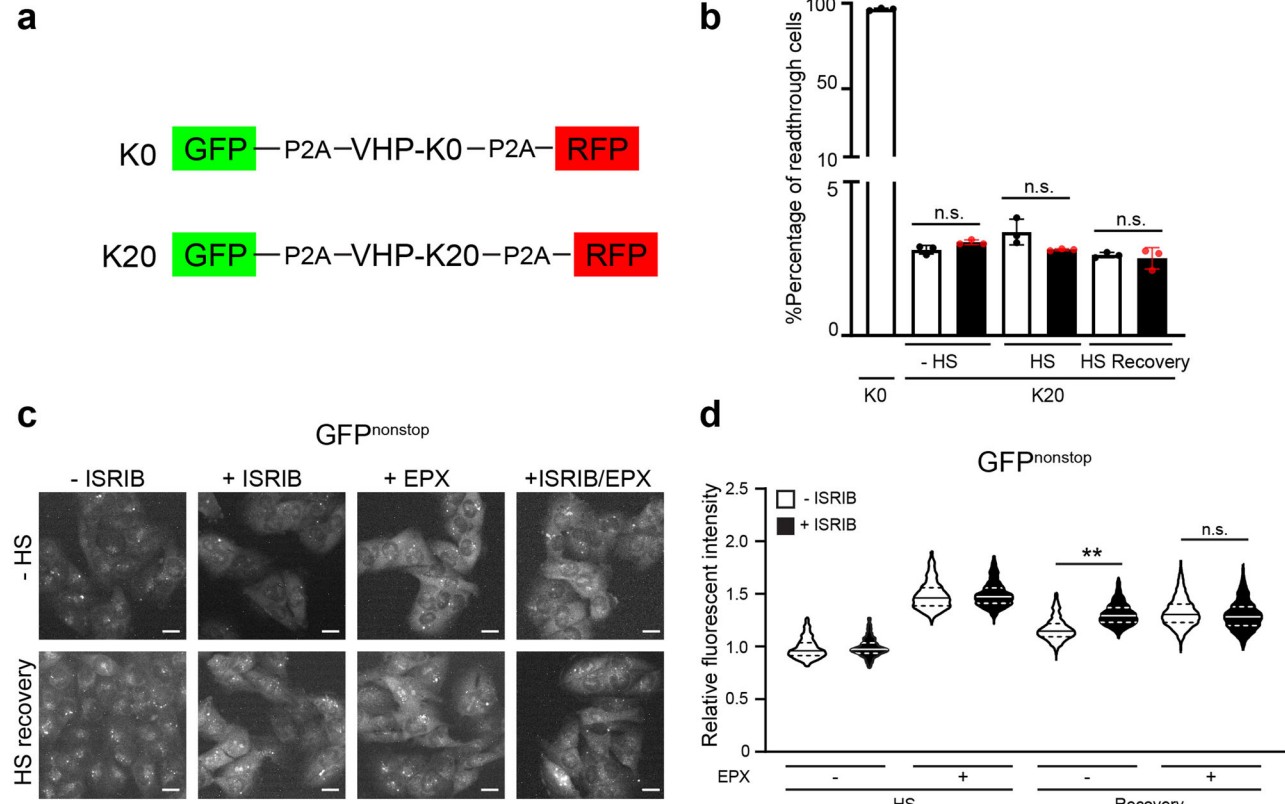

**Fig. 5 | ISRIB impairs degradation of ribosome quality control an RQC reporter substrate. a** Schematic representation of the readthrough double reporters GFP-P2A-K0-P2A-RFP (K0) and GFP-P2A-K20-P2A-RFP (K20). **b** Flow cytometric quantification of MelJuSo cells transfected with pmGFP-P2A-K20-P2A-RFP (K0) or pmGFP-P2A-K20-P2A-RFP (K20) respectively. Cells expressing GFP-K20-RFP were either pre-treated in the absence (-ISRIB) or presence (+ISRIB) of ISRIB for 30 min. After pre-treatment, cells were left untreated (−HS), or subjected to 43 °C heat shock (+HS) followed by 4 h recovery (HS recovery). The readthrough region was gated in K0 transfected cells, exhibiting linear correlation between GFP and RFP fluorescence. Data represent the mean ± SD. (three independent experiments, n.s.: not significant). **c** Effect of ISRIB on RQC substrate degradation. Representative micrographs of MelJuSo cells expressing GFP^nonstop. Cells were left untreated (-ISRIB), incubated with ISRIB for 30 min (+ISRIB), incubated with proteasome

inhibitor epoxomicin for 4 h (+EPX) or combined ISRIB with proteasome inhibitor epoxomicin (ISRIB + EPX) before being left untreated (−HS) or exposed to a 43 °C heat shock for 30 min and followed by recovery (HS recovery) for 4 h. Images were captured with wide-field, high-content microscope. Scale bar, 20 μm.
**d** Quantification of mean cellular YFP fluorescence intensities in MelJuSo cells stably expressing GFP^nonstop. Cells were pre-treated with or without ISRIB for 30 min and either left untreated (−HS) or exposed to a 43 °C heat shock for 30 min followed by 4 h (HS recovery), in presence or absence of the proteasome inhibitor epoxomicin (+EPX). Values were normalized to untreated control cells (-ISRIB, −HS). The frequency and distribution of the relative fluorescence intensities per cell are shown as violin plots. (three independent experiments, >1000 cells analyzed per condition, Kruskal–Wallis test, **P < 0.01, n.s.: not significant).

age-dependent accumulation of aggregation-prone proteins that are linked to neurodegeneration[49].

## Material and methods
### Plasmids
The GFP^nonstop expression plasmid (Addgene #226406) was generated by PCR amplifying the GFP open reading frame from EGFP-C1 using primers 5′-CGATCGACTAGTACGCGTGTTACAAATAAAGCAATAG CATCA-3′ (forward primer) and 5′-ACGCGTACTAGTCGATCGCTTG-TACAGCTCGTCCATG-3′ (reverse primer) and using NEBuilder HiFi DNA Assembly Master Mix (New England Biolabs) according to the manufacturer's instructions. The NLS-GFP-NLS-GFP-CL1 and NES-GFP-NES-GFP-CL1[31], was a gift from Dr. Ron Kopito (Stanford University). The pmGFP-P2A-K0-P2A-RFP (Addgene #105686) and pmGFP-P2A-K20-P2A-RFP (Addgene #105688) have been described previously[40].

### Cell culture and transfection
The human melanoma MelJuSo cell lines (RRID:CVCL_1403; gift from Jacques Neefjes, Leiden University) were cultured in DMEM + GlutaMAX (Life Technologies) supplemented with 10% fetal bovine serum in a at 37 °C and 5% $CO_2$. Cell lines are routinely tested for mycoplasma infection.

Generation of MelJuSo Ub-YFP stable cell lines have been described previously[28]. The MelJuSo GFP-ODC, NLS-GFP-NLS-GFP-CL1, NES-GFP-NES-GFP-CL1 and GFP^nonstop cell lines were created by transfection with corresponding GFP-ODC, NLS-GFP-NLS-GFP-CL1, NES-GFP-NES-GFP-CL1 and GFP^nonstop plasmids using Lipofectamine 3000 (Life Technologies) respectively. Clones were selected in the presence of 1.5 mg/ml G418 (Gibco). A bacterial artificial chromosome (BAC) containing the genomic sequence of the stress granule marker G3BP1, N-terminally tagged with mCherry via homologous recombination, was stably transfected into the MelJuSo Ub-YFP cell line[50].Transient transfection was performed using Lipofectamine 3000 (Life Technologies) according to the manufacturer's instructions.

### Chemical treatments
Before heat shock, the ISRIB treatment group cells were pre-treated with 200 nM ISRIB (Merck Millipore) for 30 min. Medium was changed right after heat shock, and the "recovery" phase was in ISRIB-free medium. Where indicated 100 nM epoxomicin (Sigma) was included during the recovery for 4 h. Controls for treatment with compounds contained a similar concentration of the DMSO solvent in the culture medium. To test the effect of ISRIB on sodium arsenate-induced stress, cells were first

pretreated with ISRIB as indicated above, followed by incubation with 50 μM sodium arsenate (Sigma) for 4 h. For the proteasome inhibitor titration experiments, Ub-YFP MelJuSo cells were treated with the indicated epoxomicin concentrations for 4 h. For the proteasome inhibitor sensitivity experiments, Ub-YFP MelJuSo cells were treated with the indicated epoxomicin concentrations with or without 200 nM ISRIB pre-treatment. For the GSK2606414 (Merck Millipore), cells were pre-treated for 30 min at 5 μM concentration, eventually followed by heat shock and medium change for the recovery period of 2 h.

### Immunofluorescence
MelJuSo cell lines were grown on coverslips overnight and treated as indicated. Cells were fixed using 4% paraformaldehyde (Life Technologies) for 15 min, permeabilized using 0.2% Triton X-100 in 1× phosphate-buffered saline (PBS) for 15 min, 100 nM glycine was added for 10 min, followed by 3% BSA (Sigma) in 1× PBS for 30 min. The primary antibodies were diluted in 0.1% Tween-20 in PBS and incubated overnight at 4 °C. The following primary antibodies were used: anti-puromycin (dilution 1:200; MABE343; Sigma), anti-TIA1 (dilution 1:200; #140595; Abcam). Goat anti-rabbit IgG or goat anti-mouse IgG coupled to AlexaFluor 488 or 568 (Life Technologies) were diluted 1:500 in 0.1% tween-20 in PBS. Nuclear staining was performed using Hoechst 33342 (Molecular Probes) 1:5000 in PBS for 15 min. Fixed cells were examined with a Zeiss LSM 880 confocal laser scanning microscope (Plan-Neofluar ×63/1.4 oil objective). Image processing was performed with FiJi, and quantitative analyses were performed using CellProfiler software.

### Immunoblotting
Equal amounts of cells were lysed in 1× LDS sample buffer (Life Technologies) containing 10% NuPAGE reducing agent (Life Technologies), and lysates were boiled at 95 °C for 5 min. Cell protein extracts were resolved by Bis-Tris polyacrylamide gel electrophoresis gels (Life Technologies) and run in 1× MOPS buffer (Life Technologies). Proteins were transferred onto nitrocellulose membranes (GE Healthcare) in a Tris-glycine transfer buffer containing 20% ethanol. After blocking in 1× Tris-buffered saline (TBS)/non-fat milk 5% containing 0.1% Tween-20, membranes were incubated with primary antibodies. The following antibodies were used: anti-GFP (dilution 1:5000; Abcam, ab290, PTGL #50430-2-AP), anti-GAPDH (dilution 1:5000; Abcam, ab9485), anti-β-actin (dilution 1:5000; Abcam, ab8226), anti-puromycin (dilution 1:1000; Sigma, MABE343), anti-ubiquitin (dilution 1:1000; Santa Cruz, sc-8017), anti-eIF2α(pSer51) (dilution 1:1000; Cell Signaling, #3389), eIF2α (dilution 1:1000; Cell Signaling Technology, #2103) and ATF4 (dilution 1:1000; Cell Signaling Technology #11815). After incubation with the primary antibody, the membranes were washed with TBS-Tween-20 0.1% and incubated with secondary goat anti-rabbit or anti-mouse horseradish peroxidase (HRP)-linked antibodies (dilution 1:5000). Detection was performed by enhanced chemiluminescence (Amersham ECL reagents, GE Healthcare) on Medical X-ray films (Fujifilm). Alternatively, secondary antibodies coupled to near-infrared fluorescent dyes (dilution 1:10,000; LI-COR) were used, and membranes scanned with an Odyssey scanner (LI-COR) or Bio-Rad ChemiDoc and analyzed with Image Studio Lite analysis software version 5.2 (LI-COR).

### Pulse-chase analysis
Cells were seeded overnight in a 6-well plate. Thirty minutes before thermal stress was induced, methionine was depleted with methionine-free RPMI medium (Life Technologies) simultaneously pre-treating with ISRIB. After heat shock, the medium was replaced with methionine-free RPMI medium containing 50 μM of the methionine analog L-homopropargylglycine (HPG) (Jena Bioscience, CLK-1067) for incorporation of HPG during the recovery phase. After 2 h incorporation of HPG, cells were washed with PBS and followed by 1-h HPG chase induction adding methionine-free RPMI containing 3 mM L-Methionine. After washing cells once in PBS, lysis was performed for 30 min using 250 μl/sample. After 10 min centrifugation at 4 °C and 16,000 × g, supernatant was used to perform Click chemistry for

30 min using 10 μM 800CW Azide Infrared Dye (LI-COR, CA1007-02). Dilution buffer was added to dilute the detergent concentration before the immunoprecipitation (IP) was performed with equilibrated GFP-Trap A beads by end-over-end tumble for 1.5 h at 4 °C. Collection of beads bound material and following washing steps with lysis or dilution buffer were done by centrifugation at 4 °C and 2500 × g for 2 min discarding the supernatant. Next, 2× LDS sample buffer with NuPage reducing agent was added to beads before boiling samples for 10 min at 95 °C to dissociate immune complexes. Prior proceeding with SDS-PAGE and western blotting, beads were spun down at 2500 × g for 1 min. Supernatant was loaded on the gel. In-gel detection of fluorescent labeled HPG was performed with an 800 nm laser was performed with an Odyssey scanner (LI-COR). The gel was next transferred to a nitrocellulose membrane for detection of total Ub-YFP levels by western blot analysis.

### TUBE pulldown
After indicated treatment, cells were harvested and lysed in RIPA buffer on ice for 30 min. After 10 min centrifugation at 4 °C and 16,000 × g, supernatant was mixed with pre-equilibrium TUBE beads (Life Sensors) and incubated overnight at 4 °C. The beads were washed three times with detergent free RIPA buffer. After the last wash, 2× LDS sample buffer with reducing agent was added to beads before boiling samples for 10 min at 95 °C to dissociate immune complexes. Prior to proceeding with SDS-PAGE and western blotting, beads were spun down at 2500 × g for 1 min. The supernatant was loaded on the gel.

### High-content microscopy
MelJuSo cell lines were seeded in 96-well plates suitable for imaging (Falcon or Miltenyi). For live imaging experiments, the medium was replaced with Leibovitz's L-15 medium (Life Technologies) ahead of the imaging session. Imaging was performed using an automated wide-field microscope (ImageXpress Micro; Molecular Devices), at 37 °C. For imaging experiments on fixed samples, cells were treated with 4% paraformaldehyde (Life Technologies) for 10 min, and nuclei were stained using Hoechst 33342 (Molecular Probes) 1:5000 in PBS for 30 min. Fixed samples were imaged with automated wide-field microscopes (ImageXpress Micro or ImageXpress Pico; Molecular Devices). Fluorescence intensity was quantified non-blinded with the CellProfiler software in which nuclear and cytoplasmic segmentations were determined by intensity thresholding of the Hoechst and GFP signals. Fluorescence intensities of YFP and GFP were measured in the nuclear segmentations for all experiments, except for NES-GFP-CL1, for which the cytoplasmic intensities were measured.

### Time-lapse imaging
MelJuso Ub-YFP were seeded in 12-well plates (Sarstedt) at 12,000 cells per well density and incubated overnight. The next day, cells were treated with compounds and/or heat shock, the regular medium was replaced with Leibovitz's L-15 medium (Life Technologies). If required by the experiment, further treatments with the proteasome inhibitor epoxomicin were performed at this stage. The plates were then moved to the Sartorius IncuCyte S3, which keeps cells at 37 °C, 5% CO$_2$. The cells were imaged at time intervals of 1 h, for the required number of hours. For each well, they were imaged 4 different sites. Cell confluency and Ub-YFP fluorescence intensity were quantified with the IncuCyte software (2019B Rev2 version).

### Proteasome activity
MelJuSo Ub-YFP cells were treated as indicated and harvested in lysis buffer (25 mM HEPES pH 7.2, 50 mM NaCl, 1 mM MgCl$_2$, 1 mM ATP, 1 mM DTT, 10% glycerol, 1% Triton X-100). After centrifugation, protein concentrations were measured with the protein assay dye reagent (Bio-Rad), and 10 μg proteins were mixed with 80 μl reaction buffer (lysis buffer) and 10 μl suc-LLVY-AMC (Affiniti, P802) for a final concentration of 1 μM suc-LLVY-AMC. As a control, 100 nM of the proteasome inhibitor epoxomicin (Sigma) was added to the reaction mixture. Samples were analyzed in a microplate reader (FLUOStar OPTIMA) at 355 nm/460 nm every minute for 1 h.

## Triton X-100 solubility assay

MelJuSo expressing Ub-YFP cells were seeded in 6-well plates and incubated overnight. Prior to lysis, cells were either pretreated with DMSO or ISRIB for 30 min, followed by either left untreated or exposed to 43 °C for 30 min. Cells were washed twice in cold PBS, followed by scraping in 400 µl lysis buffer on ice (1% Triton X-100 (Sigma), 1× complete EDTA-free Protease Inhibitor Cocktail (Roche), 20 mM N-Ethylmaleimide (Sigma), in PBS). Lysates were left on ice for 30 min, 200 uL of lysates were added with SDS-PAGE loading buffer and followed by 5 min of boiling at 95 °C (Input fraction). Two hundred µl of lysates were followed by centrifugation (12,000 × g, 4 °C, 10 min). SDS-PAGE loading buffer was added to the supernatants, followed by 5 min of boiling at 95 °C (Triton X-100 soluble fraction). The pellet was resuspended in a loading buffer and followed by boiling for 10 min at 95 °C (Triton X-100 insoluble fraction).

## Translation readthrough assay

MelJuSo cells were seeded in 12-well plates and incubated overnight. The cells were transfected with the pmGFP-P2A-K0-P2A-RFP (K0) or pmGFP-P2A-K20-P2A-RFP (K20) for 24 h using lipofectamine 3000. The transfected cells were treated with or without ISRIB for 30 min, and either left without treatment (−HS), heat shocked for 30 min (+HS), or heat shocked for 30 min followed by 4 h recovery (Recovery). The GFP and RFP fluorescence intensities were detected by flow cytometry (BD LSR II), and data were analyzed with the FlowJo v10 software (for gating strategy, see Suppl. Fig. 4).

## Statistical analysis

Statistical analyses were performed using GraphPad Prism version 8.3. To test for Gaussian distribution, the D'Agostino & Pearson or Shapiro-Wilk normality test (for smaller sample sizes) was used. If the normality test was passed, data were analyzed by Student's unpaired $t$ test (two groups) or by one-way analysis of variance (more than two groups). If the data were not normally distributed, statistical analysis was performed using the non-parametric Mann-Whitney test (two groups) or Kruskal–Wallis test for multiple comparisons, with Dunnett's or Tukey's test to adjust for multiple comparisons. Grubbs' test was used for the detection of outliers. Adjusted $p$ values are shown. Data are shown as mean ± SD (standard deviation), unless stated otherwise, as indicated in each figure legend. The following p-values were considered significant: $*P \leq 0.05$; $**P \leq 0.01$; $***P < 0.001$; $****P < 0.0001$.

## Reporting summary

Further information on research design is available in the Nature Portfolio Reporting Summary linked to this article.

## Data availability

Source data for graphs can be found in Suppl. Data 1. Uncropped blots can be found in Suppl. Fig. 5. Plasmids are available at Addgene (Addgene: Nico Dantuma Lab Materials). This study does not contain deposited data.

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

## Acknowledgements
We thank Vanessa Aires-Mofreita for practical help, Maria Masucci and the members of the Dantuma lab and Masucci lab for helpful input and the Biomedicum Imaging Center (BIC) and Biomedicum Flow Cytometry Core Facility for their assistance. This work was supported by the Swedish Research Council (N.P.D. 2021-02562), the Swedish Cancer Society (N.P.D. CAN 211653Pj), the Joint Program Neurodegenerative Diseases (JPND) (S.A., N.P.D.), DFG TRR 237 Nucleic Acid Immunity (S.A. Project number 369799452, the Chinese Scholarship Council (S.X.) and the Karolinska Institute (KID grant, E.B). M.E.G. was supported by research fellowships from the Deutsche Forschungsgemeinschaft (DFG) (GI-1329/1-1). N.P.D. is a member of the COST network ProteoCure.

## Author contributions
S.X., S.A., F.A.S., N.P.D. conceptualized the study; S.X., M.E.G., E.B. performed and analyzed experiments; I.P. generated stable cell lines; F.A.S. assisted in microscopy and image analysis; S.X., N.P.D. wrote the manuscript; S.A., N.P.D. coordinated the project; all authors edited and approved the final manuscript.

## Funding

## Competing interests
The authors declare no competing interests.
