## [Peer Review File · Communications Biology]

Reviewers' comments:

Reviewer #1 (Remarks to the Author):

This manuscript presents a comprehensive set of experiments showing that blunting the ISR with ISRIB causes accumulation of polyubiquitinated, aggregation-prone proteins particularly RQC substrates.

This work is novel and important and will be of interest to a broad range of scientists.

I recommend publication after the minor edits listed below.

In the figures, replace “recovery” by HS recovery” for clarity.

Lane 42: eIF2 α acts as a competitive inhibitor of eIF2B. Delete “competitive”, P-eIF2 is a non-competitive inhibitor of eIF2B as the authors explain later.

Ref 6 should be replaced by

Sidrauski, C. et al. Pharmacological brake-release of mRNA translation enhances cognitive memory. *eLife* 2, e00498 (2013).

Lane 64. Ref12 is unrelated to the point. Delete “arguing for therapeutic applications of ISRIB in a clinical setting”. There is a huge gap between mice and clinical trials.

Lane 72 Also cite

Krzyzosiak, A. et al. Target-Based Discovery of an Inhibitor of the Regulatory Phosphatase PPP1R15B. *Cell* 174, 1216-1228.e19 (2018).

Lane 168. Delete “instead”

Lane 197. Replace assumption by notion.

Lane 200. They could simplify their writing ie “these results indicate”...

Lane 307. Also cite Huntington disease and

Krzyzosiak, A. et al. Target-Based Discovery of an Inhibitor of the Regulatory Phosphatase PPP1R15B. *Cell* 174, 1216-1228.e19 (2018).

In the discussion, they may want to expand citing a recent paper that reported that ISRIB aggravates ALS in mice.

Reviewer #2 (Remarks to the Author):

The manuscript „Chemical inhibition of the integrated stress response impairs the ubiquitin-proteasome system“ by Xu and colleagues describes the effect of the integrated stress response inhibitor ISRIB on the degradation of ubiquitinated proteins and particularly the functionality of ribosome quality check (RQC).

As an extensive activation of the integrated stress response (ISR) is associated with pathological conditions such as neurodegenerative diseases, inhibition of the ISR is potentially interesting from a therapeutic point of view. Therefore, a careful evaluation of the cellular effect of ISR inhibition is a relevant undertaking.

Despite this, the study remains rather superficial and is restricted to a single stress input (heat shock) and a single inhibitor (ISRIB). As ISRIB prevents the stress-induced stall of cap-dependent translation, a clearer separation between increased translation of the reporter constructs and their degradation would be necessary. Furthermore, independent methods are required to distinguish between the effect of ISRIB on altered translation and ribosome quality check.

Specific comments to the presented data are presented in the following:

Major points:

- 1) The study is based on the assumption that heat shock serves as an effective inducer of the ISR. So far, this is only shown in Supplementary Figure S1 with a P-eIF2a western blot that does not have the best quality. It would be important to clearly demonstrate, which upstream activator part of the ISR is affected (presumably PERK), along with downstream effectors such as ATF4 and its target genes at the heat shock and at multiple time points of recovery.
- 2) Throughout their study, the authors have only used recovery times up to 4 hours. It is, however, possible that the effect of ISRIB treatment only has a transient effect on the proteasome-ubiquitin system, simply delaying its activity. Longer follow-up times would be necessary to clarify this point.
- 3) The studies in Figure 1h, Figure 2 and Figure 4 were done with fluorescence reporter proteins, whose quantification by microscopy can be challenging. An independent method (e.g. western blot) would be needed to confirm the findings. Furthermore, it would be important to test the effect of ISRIB on the ribosome quality check on endogenous proteins.
- 4) Page 6: “Pulse-chase experiments confirmed a delayed clearance of the Ub-YFP reporter in ISRIB-treated cells that recovered from thermal stress, supporting the model that the elevated steady-state levels are a consequence of slower degradation of Ub-YFP during the recovery phase (Fig. 1d)“

As ISRIB will lead to a stronger overall protein translation after heat shock compared to the controls, it is very likely that this effect prevails and Ub-YFP accumulates faster instead of being degraded slower. This needs to be addressed by independent means, e.g. the usage of protein translation inhibitors that equalize the de novo synthesis of ISRIB- and control treated cells. The same is relevant for the other figures addressing this issue.

5) Figure 3b: the nuclear localisation of the puromycin signal is not so clear. This result should be confirmed with an independent method (e.g. confocal microscopy or nuclear fractionation)

6) How is the viability of the treated cells affected by ISRIB? Does the addition of ISRIB alter the sensitivity of the cells to proteasome inhibitors such as e.g. bortezomib?

7) In order to draw a general conclusion from the study, it would be important to perform key experiments with an independent ISR inducing stress and another ISR inhibitor.

Minor points:

1) Page 3, Introduction: "...couples cellular stress detection to protein synthesis with as final goal to maintain protein homeostasis." Typo, should be: "... with the final goal to maintain protein homeostasis."

2) Page 6: "ISRIB did not have a significant effect on the enzymatic activity of proteasomes in thermally stressed cells (Fig. S4)"

 I believe that authors would like to refer to Figure S3

Reviewer #3 (Remarks to the Author):

This manuscript, authored by Shanshan Xu and co-workers, deals with the interplay between the integrated stress response (ISR) and ribosome-associated quality control (RQC) systems which responds to stalled and/or collided ribosomes. Xu et al show that a combination of heat shock and treatment of cells with the well-known ISR inhibitor ISRIB exacerbates proteotoxic stress. ISRIB does so by allowing for continued protein synthesis in the face of an already over-burdened ubiquitin-proteasome system.

The manuscript is clear and well-written and all presented data are of a high technical quality, giving clear conclusions. Overall, I think this is a very nice manuscript that delivers a concise and important message. I do however have a couple of reservations and/or suggestions for improvement.

Major issues:

- I have some issues with the experiment in Fig. 3B, and I don't think that it is immediately clear what is going on with the puromycin signal. First, the experiment would benefit from showing both puromycin signals and TIA1-positive stress granules (presumably none) in the absence of heat shock. Is the no stress +ISRIB puromycin signal predominantly nuclear? And if so, do the authors envisage a specific degradation of nuclear puromycin-tagged peptides in ISRIB-treated cells? Nuclear-cytoplasmic shuttling may also influence the recorded puromycin signals, and it would be informative to perform this experiment also in the presence of a nuclear export inhibitor such as leptomycin B.

- My main issue with the manuscript at hand is the final part with LTN1. The concept appears to be

that blocking nascent peptide release via RQC partly alleviates the cellular concentration of proteins to be degraded by the proteasome. This is interesting indeed, but I think the experimental evidence is too thin and that more can/should be done. 1) more LTN1 siRNAs, 2) rescue experiments, 3) knocking down additional well-established RQC components such as NEMF and ZNF598, just to mention some suggestions for improvement. Especially NEMF depletion could be interesting for the following reason: LTN1 depletion is expected to lead to increased NEMF-dependent CAT tailing, and while release of nascent chains would be inhibited, these might also aggregate in the context of ribosomes, confusing the result. NEMF knockdown on the other hand would curtail CAT-tailing and might therefore have a different impact on the Ub-YFP signal in Figure 3H.

- The story is mainly developed by using ISRIB as a tool to inhibit the ISR and inferring general conclusions on the impact of this pathway. However, ISRIB is still “just” a chemical compound with likely pleiotropic and off-target effects. It would help the story if some basic conclusion(s) could be supported by alternative strategies. An example would be to determine the relevant eif2-alpha kinase(s) (GCN2? PERK?) that can either be inhibited or knocked out.

Minor issues:

- In a couple of places, independent loading controls should be added to western blot experiments – in particular Figs. 1D and 3C.

- While it is “straightforward” to make a nuclear mask for image analysis based on a DNA dye, cytoplasmic segmentation is no easy task. In Fig. 2C, the authors measure the uneven cytoplasmic GFP-signal based on a Cell Profiler script, not with a cytoplasmic marker. It should be clarified in detail how this segmentation is performed and how well the method is able to segment the cytoplasmic compartments.

- In figure 3A, a key control with -heat shock and +ISRIB appears to be missing.

Rebuttal COMMSBIO-23-4474 “Chemical inhibition of the integrated stress response impairs the ubiquitin-proteasome system”

We would like to thank the reviewers for their constructive comments. We feel that addressing the concerns of the reviewers has considerably strengthened our study. Additional experiments have been included in the manuscript and where requested we have edited the text and clarified issues that were unclear.

Reviewer #1

Reviewer #1: “This manuscript presents a comprehensive set of experiments showing that blunting the ISR with ISRIB causes accumulation of polyubiquitinated, aggregation-prone proteins particularly RQC substrates. This work is novel and important and will be of interest to a broad range of scientists. I recommend publication after the minor edits listed below.”

Authors: We thank the reviewer for acknowledging the novelty and importance of our work as well as the constructive feedback

Reviewer #1: “In the figures, replace “recovery” by HS recovery” for clarity.”

Authors: We have changed this accordingly throughout the manuscript text and figures.

Reviewer #1:” Lane 42: eIF2 α acts as a competitive inhibitor of eIF2B. Delete “competitive”, P-eIF2 is a non-competitive inhibitor of eIF2B as the authors explain later”

Authors: This has been rephrased. See lines 39-41

Reviewer #1: “Ref 6 should be replace by Sidrauski, C. et al. Pharmacological brake-release of mRNA translation enhances cognitive memory. eLife 2, e00498 (2013).”

Authors: This citation has been replaced. See line 41.

Reviewer #1: “Lane 64. Ref12 is unrelated to the point. Delete “arguing for therapeutic applications of ISRIB in a clinical setting”. There is a huge gap between mice and clinical trials.”

Authors: This has been removed.

Reviewer #1: “Lane 72 Also cite Krzyzosiak, A. et al. Target-Based Discovery of an Inhibitor of the Regulatory Phosphatase PPP1R15B. Cell 174, 1216-1228.e19 (2018).”

Authors: This citation has been included. See line 71.

Reviewer #1: “Lane 168. Delete “instead”. Lane 197. Replace assumption by notion. Lane 200. They could simplify their writing ie “these results indicate”...”

Authors: These changes have been made. See line 221.

Reviewer #1: “Lane 307. Also cite Huntington disease and Krzyzosiak, A. et al. Target-Based Discovery of an Inhibitor of the Regulatory Phosphatase PPP1R15B. Cell 174, 1216-1228.e19 (2018).”

Authors: Huntington’s disease and citation have been included. See lines 302-306.

Reviewer #1: “In the discussion, they may want to expand citing a recent paper that reported that ISRIB aggravates ALS in mice.”

Authors: We mention the recently reported effects of ISRIB on ALS in mouse models and included additional citations. This issue is complicated as both worsening and improvement of the pathology has been reported. See lines 300-302.

Reviewer #2

Reviewer #2: “The manuscript “Chemical inhibition of the integrated stress response impairs the ubiquitin-proteasome system” by Xu and colleagues describes the effect of the integrated stress response inhibitor ISRIB on the degradation of ubiquitinated proteins and particularly the functionality of ribosome quality check (RQC).

As an extensive activation of the integrated stress response (ISR) is associated with pathological conditions such as neurodegenerative diseases, inhibition of the ISR is potentially interesting from a therapeutic point of view. Therefore, a careful evaluation of the cellular effect of ISR inhibition is a relevant undertaking.

Despite this, the study remains rather superficial and is restricted to a single stress input (heat shock) and a single inhibitor (ISRIB). As ISRIB prevents the stress-induced stall of cap-dependent translation, a clearer separation between increased translation of the reporter constructs and their degradation would be necessary. Furthermore, independent methods are required to distinguish between the effect of ISRIB on altered translation and ribosome quality check.”

Authors: We thank the reviewer for the helpful input and appreciate the concerns regarding the used stress paradigm and use of single inhibitor which are addressed below.

Reviewer #2: “Major points: 1) The study is based on the assumption that heat shock serves as an effective inducer of the ISR. So far, this is only shown in Supplementary Figure 1 with a P-eIF2a western blot that does not have the best quality. It would be important to clearly demonstrate, which upstream activator part of the ISR is affected (presumably PERK), along with downstream effectors such as ATF4 and its target genes at the heat shock and at multiple time points of recovery.”

Authors: The heat shock paradigm was chosen because of its reversible nature as well as the fact that activation of the ISR by heat shock is well documented in the literature. Indeed, it has been shown that heat shock activates PERK, as the reviewer proposed. See for example (Bettaieb & Averill-Bates, 2015; Elvira et al, 2020; Park et al, 2018). These papers are now cited in the manuscript. Accordingly, we found that heat shock induces expression of ATF4 and phosphorylation of eIF2 α (see Suppl. Fig. 1a,b). Moreover, inhibition of PERK by GSK2606414 had a similar effect on the UPS in thermally stressed cells as ISRIB consistent with a prominent role of this branch of the ISR in cells recovering from heat shock (see new Fig. 2a). See lines 96-98, 106-108, Suppl. Fig. 1a,d and Fig. 2a, citations 23-25.

Reviewer #2: “2) Throughout their study, the authors have only used recovery times up to 4 hours. It is, however, possible that the effect of ISRIB treatment only has a transient effect on the proteasome-ubiquitin system, simply delaying its activity. Longer follow-up times would be necessary to clarify this point.”

Authors: We have included a time kinetics following the levels of the Ub-YFP reporter up to 14 hours in thermally stressed cells that were left untreated or exposed to ISRIB (See new Fig. 1c). This shows that the Ub-YFP reporter levels accumulate in ISRIB-treated cells up to 4-5 hours after

the heat shock upon which they start to decline reaching basal levels. This also shows that the effect of ISRIB on the UPS during the recovery phase is transient. This is explained in the result section. See lines 125-129 and Fig. 1c.

Reviewer #2: “3) The studies in Figure 1h, Figure 2 and Figure 4 were done with fluorescence reporter proteins, whose quantification by microscopy can be challenging. An independent method (e.g. western blot) would be needed to confirm the findings. Furthermore, it would be important to test the effect of ISRIB on the ribosome quality check on endogenous proteins.”

Authors: We have included a Western blot confirming the increase in Ub-YFP steady-state levels in ISRIB cells recovering from heat shock (see new Fig. 1d). We would like to note that Western blotting is by nature a semiquantitative analysis as there is not necessarily a linear correlation between the detected band intensity and the levels of the protein. DRiPs are a pool of endogenous RQC proteins. We show in Fig. 4 that the pools of ubiquitylated and insoluble DRiPs are increased in ISRIB-treated, thermally stressed cells. See lines 125-129, Fig. 1d and Fig. 4c,d.

Reviewer #2: “4) Page 6: “Pulse-chase experiments confirmed a delayed clearance of the Ub-YFP reporter in ISRIB-treated cells that recovered from thermal stress, supporting the model that the elevated steady-state levels are a consequence of slower degradation of Ub-YFP during the recovery phase (Fig. 1d)” As ISRIB will lead to a stronger overall protein translation after heat shock compared to the controls, it is very likely that this effect prevails and Ub-YFP accumulates faster instead of being degraded slower. This needs to be addressed by independent means, e.g. the usage of protein translation inhibitors that equalize the de novo synthesis of ISRIB- and control treated cells. The same is relevant for the other figures addressing this issue.”

Authors: The reviewer is correct that we cannot exclude that the persistent synthesis of Ub-YFP in ISRIB-treated cells contribute to the elevated levels observed during the recovery phase. Having said that, the labeling experiment in Figure 1e shows very similar levels of Ub-YFP directly after the pulse labeling, suggesting that synthesis of the Ub-YFP has been comparable during heat shock. The same experiment also shows that the degradation of the labelled Ub-YFP is delayed in ISRIB-treated cells. Together this strongly suggests that delayed degradation is primarily responsible for the increase in Ub-YFP levels. We acknowledge in the revision the fact that we cannot exclude a minor contribution of the differences in protein synthesis even though our data clearly show that retarded degradation is primary responsible for the increase in Ub-YFP levels. See lines 133-137.

Reviewer #2: “5) Figure 3b: the nuclear localisation of the puromycin signal is not so clear. This result should be confirmed with an independent method (e.g. confocal microscopy or nuclear fractionation)”

Authors: We are wondering if there could have been a problem with the PDF conversion as in the original figure the nuclear localization of the puromycin signal is quite clear.

Reviewer #2: “6) How is the viability of the treated cells affected by ISRIB? Does the addition of ISRIB alter the sensitivity of the cells to proteasome inhibitors such as e.g. bortezomib?”

Authors: We have followed proliferation of cells recovering from heat shock in the absence or presence of ISRIB (see new Suppl. Fig. 1c). In addition, we have treated the cells with an increasing concentration of the highly specific inhibitor epoxomicin (see new Suppl. Fig. 3b). This did not reveal any detectable effect of ISRIB on cell viability nor a detectable increase in their sensitivity to epoxomicin. See lines 122-124 and Suppl. Fig. 1c and lines 153-155 and Suppl. Fig. 3b.

Reviewer #2: “7) In order to draw a general conclusion from the study, it would be important to perform key experiments with an independent ISR inducing stress and another ISR inhibitor.”

Authors: In Figure 2 we show that ISRIB has a similar effect on cells that have been exposed to sodium arsenate causing oxidative stress. We also show in the same figure that the GSK2606414, which inhibits phosphorylation of PERK and acts upstream of eIF2 α phosphorylation, has a similar effect on the UPS as ISRIB. Thus, both an independent inducer of ISR and an independent ISR inhibitor confirm our results. See lines 158-166 and Fig. 2a, lines 167-75 and Fig. 2b-d.

Reviewer #2: “Minor points: 1) Page 3, Introduction: “...couples cellular stress detection to protein synthesis with as final goal to maintain protein homeostasis.” Typo, should be: “... with the final goal to maintain protein homeostasis.” 2) Page 6: “ISRIB did not have a significant effect on the enzymatic activity of proteasomes in thermally stressed cells (Fig. S4)””

Authors: We thank the reviewer for bringing these mistakes to our attention. This has been corrected.

Reviewer #3

Reviewer #3: “This manuscript, authored by Shanshan Xu and co-workers, deals with the interplay between the integrated stress response (ISR) and ribosome-associated quality control (RQC) systems which responds to stalled and/or collided ribosomes. Xu et al show that a combination of heat shock and treatment of cells with the well-known ISR inhibitor ISRIB exacerbates proteotoxic stress. ISRIB does so by allowing for continued protein synthesis in the face of an already over-burdened ubiquitin-proteasome system.

The manuscript is clear and well-written and all presented data are of a high technical quality, giving clear conclusions. Overall, I think this is a very nice manuscript that delivers a concise and important message. I do however have a couple of reservations and/or suggestions for improvement.”

Authors: We are pleased to hear that the reviewer finds the data of high quality and supports our conclusions. We thank the reviewer for the helpful insights.

Reviewer #3: “Major issues: - I have some issues with the experiment in Fig. 3B, and I don’t think that it is immediately clear what is going on with the puromycin signal. First, the experiment would benefit from showing both puromycin signals and TIA1-positive stress granules (presumably none) in the absence of heat shock. Is the no stress +ISRB puromycin signal predominantly nuclear? And if so, do the authors envisage a specific degradation of nuclear puromycin-tagged peptides in ISRIB-treated cells? Nuclear-cytoplasmic shuttling may also influence the recorded puromycin signals, and it would be informative to perform this experiment also in the presence of a nuclear export inhibitor such as leptomycin B.”

Authors: The observation that puromycin-labelled DRiPs are enriched in the nuclear compartment is indeed intriguing. Two recent studies (one of them from our group) has studied this phenomenon in more detail (Mediani et al, 2019; Xu et al, 2022). These studies show that they reach the nuclear compartment through diffusion and are not dependent on active import. For readers who are interested in this phenomenon, we refer now to these studies. While intriguing, we feel that a more detailed analysis lies outside the scope of this study. See lines 211-214.

Reviewer #3: “- My main issue with the manuscript at hand is the final part with LTN1. The concept appears to be that blocking nascent peptide release via RQC partly alleviates the cellular

concentration of proteins to be degraded by the proteasome. This is interesting indeed, but I think the experimental evidence is too thin and that more can/should be done. 1) more LTN1 siRNAs, 2) rescue experiments, 3) knocking down additional well-established RQC components such as NEMF and ZNF598, just to mention some suggestions for improvement. Especially NEMF depletion could be interesting for the following reason: LTN1 depletion is expected to lead to increased NEMF-dependent CAT tailing, and while release of nascent chains would be inhibited, these might also aggregate in the context of ribosomes, confusing the result. NEMF knockdown on the other hand would curtail CAT-tailing and might therefore have a different impact on the Ub-YFP signal in Figure 3H.”

Authors: We thank the reviewer for raising this issue as it turned out to be very relevant input during the review process. We have put a considerable amount of effort into addressing this point, which is also the main reason for the extensive time it took us to prepare the revision.

Unfortunately, we have been unable to obtain convincing additional data supporting a role for RQC. We have tested multiple siRNAs directed against LTN1, ZNF589 and NEMF. For the LTN1 and ZNF589, the different siRNAs gave disparate results, while siRNAs directed against NEMF reduced the ISRIB-induced accumulation of Ub-YFP. However, control experiments suggested that, for reasons we do not understand, NEMF depletion appears to inhibit the synthesis of Ub-YFP, which complicated the interpretation of the data. With these new data at hand, we feel that the experimental work on a role of RQC is inconclusive. We have therefore removed the LTN1 data and refrained from speculating on the role of LTN1-targeted ubiquitylation in the effect of ISRIB. This detail should not distract, however, from our main conclusion that inhibition of the ISR impairs UPS functionality in stressed cells and inhibits the degradation of RQC and soluble substrates in the cytosolic compartment.

Reviewer #3: “- The story is mainly developed by using ISRIB as a tool to inhibit the ISR and inferring general conclusions on the impact of this pathway. However, ISRIB is still “just” a chemical compound with likely pleiotropic and off-target effects. It would help the story if some basic conclusion(s) could be supported by alternative strategies. An example would be to determine the relevant eIF2-alpha kinase(s) (GCN2? PERK?) that can either be inhibited or knocked out.”

Authors: We have included additional data that show GSK2606414, which inhibits phosphorylation of PERK and acts upstream of eIF2 α phosphorylation, has a similar effect on the UPS as ISRIB (Fig. 2a). See lines 258-166 and Fig. 2a.

Citations

Bettaieb A, Averill-Bates DA (2015) Thermotolerance induced at a mild temperature of 40 degrees C alleviates heat shock-induced ER stress and apoptosis in HeLa cells. *Biochim Biophys Acta* **1853**: 52-62

Elvira R, Cha SJ, Noh GM, Kim K, Han J (2020) PERK-Mediated eIF2alpha Phosphorylation Contributes to The Protection of Dopaminergic Neurons from Chronic Heat Stress in Drosophila. *Int J Mol Sci* **21**

Mediani L, Guillen-Boixet J, Vinet J, Franzmann TM, Bigi I, Mateju D, Carra AD, Morelli FF, Tiago T, Poser I, Alberti S, Carra S (2019) Defective ribosomal products challenge nuclear function by impairing nuclear condensate dynamics and immobilizing ubiquitin. *EMBO J* **38**: e101341

Park S, Lim Y, Lee D, Elvira R, Lee JM, Lee MR, Han J (2018) Modulation of Protein Synthesis by eIF2alpha Phosphorylation Protects Cell from Heat Stress-Mediated Apoptosis. *Cells* **7**

Xu et al, Rebuttal letter

Xu S, Gierisch ME, Schellhaus AK, Poser I, Alberti S, Salomons FA, Dantuma NP (2022) Cytosolic stress granules relieve the ubiquitin-proteasome system in the nuclear compartment. *EMBO J*: e111802

REVIEWERS' COMMENTS:

Reviewer #2 (Remarks to the Author):

In their revised version of the manuscript „Chemical inhibition of the integrated stress response impairs the ubiquitin-proteasome system“, the authors addressed all points raised by the reviewers. Using additional data and the requested controls, they could further substantiate their main findings. Issues that were not quite clear, such as the involvement of the E3 ubiquitin ligase listerin and RQC in ISRIB-mediated UPS, was deleted from the paper.

Altogether, this is now a sound and very interesting manuscript.

Reviewer #3 (Remarks to the Author):

The authors have revised their manuscript without performing too many new experiments. Still, I feel that my concerns have been adequately addressed. I especially condone their decision to remove the LTN1 data in the absence of further experimental evidence for the original concept.